# Neural Network-Driven Accuracy Enhancement for Wearable Eye-Tracking Devices Using Distance Information

**Anonymous X**                                                    oooo.oooo@oooo.ooo

**Editor:** Editor's name

## Abstract

Eye-tracking devices are convenient for interpreting human behaviours and intentions, opening the way to contactless human-computer interaction for various application domains. Recent evolutions have enhanced them into wearable eye-tracking devices that opened the technology to the real world by allowing wearers to move freely and use them in regular indoor or outdoor activities. However, the gaze estimate from wearable devices remains more approximative than standard stationary eye-tracking devices due to their design constraints and a lack of interpretation of the three-dimensional scene of their wearer. This paper proposes to improve the gaze estimation accuracy of wearable eye-tracking devices using a framework that involves two neural networks, *CorNN* and *CalNN*. The *CorNN* corrects the bias induced by the distance between the observer and the gaze locations, primarily due to the parallax and lens distortion effects. While the *CalNN* is used to improve wearer-specific calibration. A robotic data collection system is implemented to automate training data acquisition for these networks. The proposed network has been demonstrated over a Pupil Labs Invisible eye-tracking device and tested on 11 wearers, showing improvement in the average gaze estimation accuracy on all wearers, especially at short-range reads.

**Keywords:** Head-mounted eye-tracking device, Neural Network-based Parallax Correction, Accuracy Improvement, Pupil Labs Invisible

## 1. Introduction

Eye gaze tracking has become a vital tool for contactless human-computer interaction. The recent applications of eye-tracking technology have extended into various areas, including robot-human guidance in industrial settings Shen et al. (2023); Berg et al. (2019); Di Maio et al. (2021); Chadalavada et al. (2020); Kratzer et al. (2020), driving monitoring systems Čegovnik et al. (2018); Xu et al. (2018); Akshay et al. (2021), and enhancing the quality of automated medical image segmentation and analysis by assessing visual attention Khosravan et al. (2017); Wang et al. (2023, 2022); Ma et al. (2023). Wearable eye-tracking devices have extended the technology to many conditions, reduced the constraints of use (e.g. they do not require any chin-rest), and simplified their calibration protocol (e.g. the Pupil Labs Invisible only needs a wearer-specific offset).

Wearable eye-tracking devices, such as the Pupil Labs Invisible Tonsen et al. (2020) as illustrated in Figure 1, have especially received much attention due to their portability and versatility. However, challenges remain in developing accurate wearable eye-tracking devices. In particular, adapting the device to the gaze characteristics inherent to different wearers is imperative. Pupil Labs Invisible uses a camera scene offset to correct for bias. The

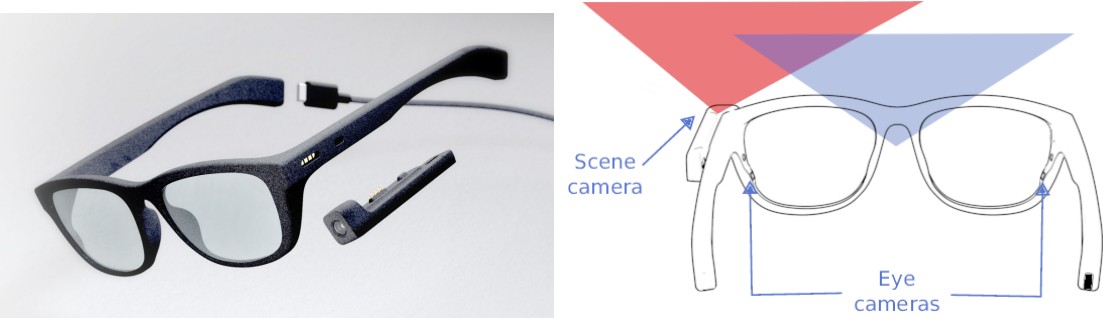

Figure 1: Pupil Labs Invisible eye tracking device. The red and blue triangles draw the camera and wearer's points of view, respectively. (https://pupil-labs.com/products/invisible/)

wearer manually sets his/her specific offset, which is added to the estimated gaze location to approximate the intended gaze location. However, this oversimplified correction cannot reflect errors due to scene camera lens distortion and parallax effects that impact the gaze estimation according to distance of the wearer with the gazed location.

This study proposes a correction framework for head-mounted eye-tracking devices using neural networks. Figure 2 illustrates the outline of the proposed framework. We used a neural network, *CorNN*, to correct the parallax and lens distortion effect. We used another network, *CalNN*, to perform wearer-specific calibration. To automate training data acquisition for these networks, we implemented a robotic data collection system with the UR5e manipulator (see Fig. 3). To validate the correction framework using neural networks, we collected gaze data from 11 participants and evaluated the gaze estimation accuracy. The proposed method improves all wearers' average gaze estimation accuracy compared to the baseline.

## 2. Challenges in Gaze Estimation with Eye Trackers

*Eye trackers* are devices capable of estimating the gaze location of the users by tracking their eye movements. The estimated gaze location is generally represented as pixel coordinates within a monitor screen or scene camera image. Head-stabilized and remote eye-tracking devices, such as the EyeLink 3000+ and the Tobii Pro X2-30, are widely adopted in research and are known for their remarkable gaze estimation accuracy. However, it is also known that these products require meticulous calibrations and involve constraints on the user's position to achieve this level of accuracy. These products are not portable and must be mounted at a fixed location.

In contrast, wearable *head-mounted eye trackers* allow their wearers to use them while moving freely in regular indoor and outdoor activities Cognolato et al. (2018); Franchak and Yu (2022). An example is the *Pupil Labs Invisible eye-tracker Tonsen et al. (2020)*. It looks like a pair of regular eyeglasses (see Fig.1), but with two *eye cameras* installed in the frame pointed towards the wearer's eyes, capturing the pupil centre location and corneal

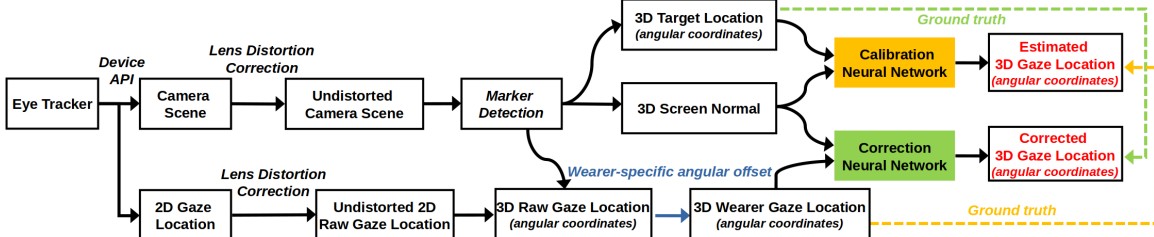

Figure 2: The proposed framework for wearable eye-tracker correction. The Eye-tracker's scene camera image and 2D gaze location are collected using the device's API. The camera lens distortion is compensated on both the 2D images and corresponding 2D gaze locations. The detected 3D marker's position and screen normal were used for network inputs to correct the 3D gaze location. Green elements are part of the Gaze Correction Neural Network. Orange elements are part of the Calibration Neural Network. Dashed arrows show the data used as the ground truth output for the corresponding networks during training. The 3D Target location is present for training and calibration purposes only.

reflections. This allows for estimating the wearer's gaze location relative to the device. Additionally, a *scene camera* is mounted on the left side of the frame, capturing the scene seen by the wearer and enabling the device's self-localisation in the wearer's environment. The wearer's gaze location in the environment can thus be determined using the eye and scene cameras.

However, design constraints in hardware make it difficult to accurately estimate the wearer's gaze location. For example, the centre of the scene camera in the Pupil Labs Invisible eye tracker is located approximately 70 mm to the left and 10 mm above the centre of the wearer's eye, resulting in a discrepancy between the scene camera's field of view and the wearer's field of view. This visual effect implied by these shifted points of view between two observers is known as the *parallax effect*. Moreover, when the parallax effect combines with camera lens distortion, the gaze estimation exhibits a spinning effect around the actual gaze location, especially when the distance is less than two meters. To address these challenges, the proposed framework in this paper utilizes a neural network to correct this spinning effect.

## 3. Method

The eye-tracking device used in our study is the Pupil Labs Invisible Tonsen et al. (2020) (Fig. 1), a lightweight wearable device that requires only minimal calibration. The device allows a wearer-specific offset to be added to the x- and y-coordinates of the 2D raw gaze location estimates, correcting biases inherent to individual wearers. This offset is manually configurable via the Pupil Labs Invisible Companion application installed on the smartphone connected to the eye-tracking device.

Consider the scenario where a person wearing the eye tracker gazes at a specific location on a monitor screen. The proposed framework, outlined in Figure 2, embeds the two-

dimensional raw gaze estimation from the camera scene image into the 3D real-world scene of the wearer. In this 3D scene, as illustrated in Figure 3, we define $g$ as the gaze location estimated by the eye tracking device projected on the screen, $n$ as the normal direction of the monitor screen plane, and $t$ as the wearer's actual gaze location on the screen (ground truth for $g$). These quantities are measured in a three-dimensional Cartesian coordinate system, with the origin being the centre of the scene camera and the $z$ axis aligned with the normal of the scene camera.

Our approach involves two neural networks: a Correction Neural Network (*CorNN*) and a Calibration Neural Network(*CalNN*). The *CorNN* uses gaze estimation ($g$) and screen orientation ($n$) as inputs to predict the actual gaze location ($t$). It is directly responsible for correcting the gaze estimation error induced by the parallax effect and the scene camera lens distortion. The *CalNN* is the reciprocal of *CorNN*, which predicts $g$ from $t$ and $n$. It is used to calculate *wearer-specific offsets*, an input parameter required by *CorNN* that captures the estimation bias inherent to individual wearers. The details of this framework are described in the following subsections.

## 3.1. Data collection

To train and test the neural networks, we acquire gaze estimations $g_i$ from a set of `fixation points` characterized by gaze target locations $t_i$ (ground truth for $g_i$) and screen orientations $n_i$. We developed a graphical interface for data acquisition, as shown in Figure 3.

The graphical interface displayed four ArUco markers at each corner of the screen and a visual target. The 3D location of the screen was estimated using the four ArUco markers, while the wearers were instructed to focus on the centre of a visual target during the data collection. The visual target was a ring-shaped animated object, consisting of two rings, a cross, and a black dot in the centre. The cross rotated continuously to facilitate visual attention to the central point. The dot and the inner ring were resizable, and we adjusted their sizes to be distinctly visible at the observing distance. We used the realtime-network-api Prietz et al. (2023) provided by Pupil Labs to access the data from the eye tracker, including the scene camera view and a 2D raw gaze estimation in the scene camera coordinates (in pixels). While collecting data, the wearer-specific offset on the Invisible companion device remained at (0, 0), its default setting.

The training dataset was established on a single-wearer recording, referred to as `Wearer 0`. For this recording, the wearer's head was stabilised on a chin-rest, and the monitor screen was mounted on a UR5e robot arm facing the wearer (see Fig. 3). The robot arm was programmed to move the screen through a regular grid of 3D way-points sequentially, generating various `fixation points` within the field of view of the scene camera. Upon arriving at each location within the grid, the robot arm remains stationary to record a new `fixation point`. The eye-tracking device's estimated 2D and 3D gaze locations and the eventual wearer's head moves were monitored during recording. When the wearer's head and gaze were observed to have stabilized according to the latest device output, we recorded an estimated 3D gaze location $g$[1], the actual target location $t$, and the camera's relative orientation to the screen $n$. Upon the completion of this recording, the robot arm moved the

---

1. $g$ was obtained by offsetting the latest gaze estimation using the average difference between the device's outputs and the actual target locations over a window of several samples, excluding outliers.

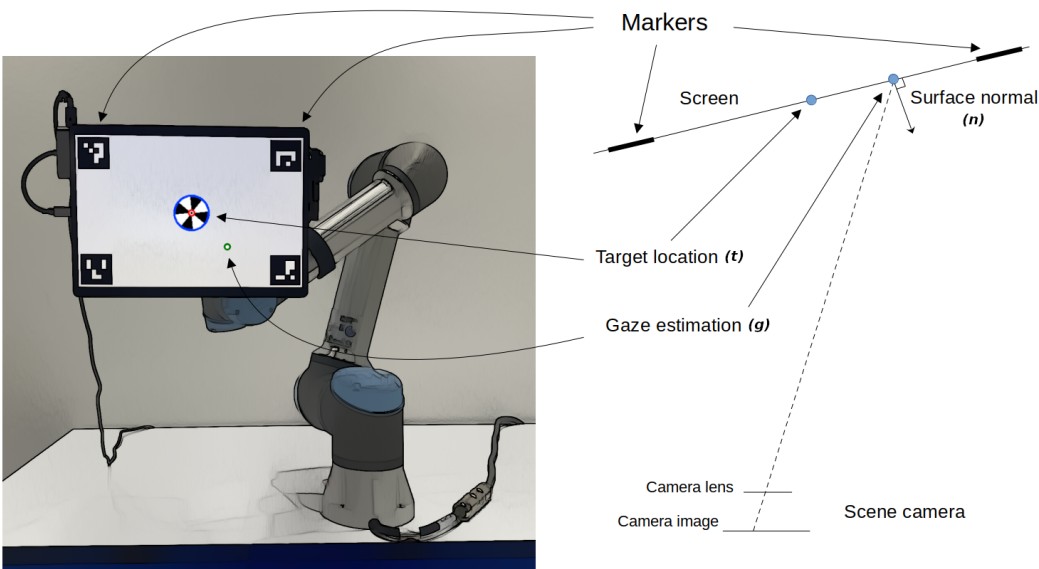

Figure 3: Actual gaze location vs. gaze estimation. This figure illustrates the screen presented to the eye tracker's wearers and the schematic of the 3D gaze estimation and scene pose reconstruction during a record. Left, the application screen with four ArUco markers, the visual target, and the gaze estimated by the eye tracking device. Right, a top view of the scene with the 3D locations of the screen and the projection for the gaze estimation onto the screen plane.

screen to the next grid point. Using the robot arm, 10405 samples were recorded within a bounding box about 1.5m wide, 1.2m high, and 1.5m deep, centred along the scene camera axis at about 500mm from the wearer.

## 3.2. Pre-processing training data

Collected data are pre-processed before training the neural networks through the following steps.

### 3.2.1. CONVERTING CARTESIAN COORDINATES TO ANGULAR COORDINATES

Let the $p$ be either $g$ (gaze estimate) or $t$ (target location). The first step of pre-processing is to convert $p$ into angular coordinates $\dot{p} = (\theta, \phi, d)$, where $\theta$ and $\phi$ as the angles between $p$ and the normal of the camera ($z$ axis) along the axes $x$ and $y$, respectively, and $d$ as the distance between $p$ and the camera's nodal point. This conversion is necessary since the error tolerance of eye-tracking devices is associated with angles.

### 3.2.2. AUGMENTING THE DATA SET

We introduce an augmentation technique to our training data set that artificially generates gaze estimation values with various potential screen orientations. The idea is as follows: Suppose a `fixation point` $(g, t, n)$ is recorded from the wearer, where $g$ becomes $\dot{g}$ after

conversion to angular coordinates. Consider the case where screen orientation $n^+$ differs from $n$, while the target $t$ remains at the same location. The eye-tracking device would return a gaze estimation $g^+$, whose angular representation $\dot{g}^+$ have the same $\phi$ and $\theta$ values as $\dot{g}$ but a different distance $d^+$. Starting from this idea, for each $t$ we generate 201 simulated $n^+$ values such that their angles with line $(O, t)$ remain below $60°$, where $O$ is the centre of the camera $O = (0, 0, 0)$. For each $n^+$, we compute its corresponding $g^+$, such that the $\phi$ and $\theta$ components of its angular representation are the same as $\dot{g}$. Only its component $d$ is modified so that $g^+$ lies on the plane $(t, n^+)$.

### 3.2.3. ADDING THE WEARER-SPECIFIC OFFSET

The gaze estimation in angular coordinates $\dot{g}_i$ needs adjustment through the *wearer-specific offset* to account for the biases inherent to individual wearers. We define the wearer-specific offset $\rho$ as an angular shift along the x-axis and y-axis of the camera, $(\theta, \phi)$. This offset is a weighted average of the difference between $\dot{t}_i$ and $\dot{g}_i$, calculated according to equation (1),

$$\rho = (\rho_\theta, \rho_\phi) = \frac{\sum_i \omega_i (\dot{t}_{i_\theta} - \dot{g}_{i_\theta}, \dot{t}_{i_\phi} - \dot{g}_{i_\phi})}{\sum_i \omega_i}, \tag{1}$$

where $\omega_i$ are weights that prioritise records closer to a defined point $C = (0, 0, 2000)$ situated 2 metres from the device, as suggested by the manufacturer.

$$\omega_i = \left(1 - \frac{\|t_i - C\|}{\max_j \|t_j - C\|}\right)^2 \tag{2}$$

We then compute the shifted gaze estimation set $\{\dot{g}'_i\} = \{(\theta'_i, \phi'_i, d'_i)\}$ by adding $(\rho_\theta, \rho_\phi)$ to the gaze estimation in angular coordinates $\dot{g}_i = (\theta_i, \phi_i, d_i)$ and re-projecting onto the screen plane,

$$\dot{g}'_i = (\theta'_i, \phi'_i, d'_i) = (\theta_i + \rho_\theta, \phi_i + \rho_\phi, d'_i) \tag{3}$$

where $d'_i$ is calculated so that $\dot{g}'_i$ lies on the screen plane $(t, n)$.

**Remark 1** *The major issue of calculating the offset according to equations (1) and (2) is that the offset puts much more weight on samples close to the manufacturer-specified location $C = (0, 0, 2000)$ than the remaining samples. However, later on, we want our calibration process to equally consider samples from locations of different distances rather than overly focusing on a pre-determined location. Therefore, when adjusting our framework to the testing wearers, we use CalNN to calculate wearer-specific offset that does not bias towards any specific location(see Section 3.4).*

### 3.3. Networks Structure and Training

The *CorNN* and *CalNN* have identical network architectures but are trained independently. The neural networks comprise 20 fully connected layers with 500 hidden units. Each fully connected layer is followed by a leaky Rectified linear unit (ReLU) and dropout layer Gal and Ghahramani (2016). We applied a dropout rate of 0.3 to obtain robustness against noisy measurements. A residual connection between the gaze coordinates and the neural network output was used He et al. (2016). The residual connection of angular coordinate

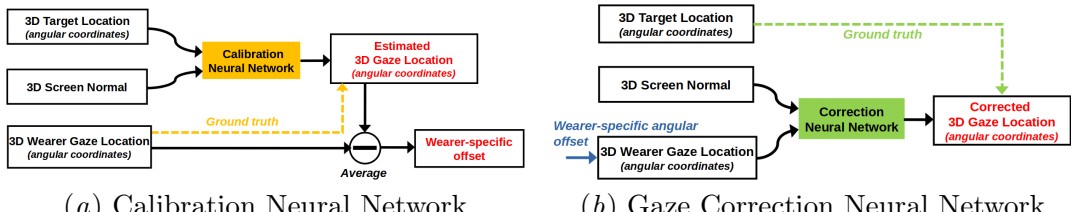

$(a)$ Calibration Neural Network $\qquad$ $(b)$ Gaze Correction Neural Network

Figure 4: Gaze Calibration and Correction Networks (Dashed arrows indicate the ground truth in training).

data provides robust correction by preventing angular warping. *CorNN* takes $(n, \dot{g}')$ as input and is trained to predict $\dot{t}$, while *CalNN* takes $(n, \dot{t})$ as input and predicts $\dot{g}'$. We use the mean squared error MSE (Eq. (4) and (5)) as the loss function for training, as follows:

$$L_{CorNN}(n, \dot{g}') = \sum_i (\dot{t}_i - CorNN(n_i, \dot{g}'_i))^2, \tag{4}$$

$$L_{CalNN}(n, \dot{t}) = \sum_i (\dot{g}'_i - CalNN(n_i, \dot{t}_i))^2, \tag{5}$$

where $L_{CorNN}$ and $L_{CalNN}$ are losses of *CorNN* and *CalNN*, respectively. The neural networks were implemented in Python 3.7 using Keras 2.11.0. Training, validation, and testing were performed on a 12GB memory GTX Titan Xp workstation (NVIDIA Santa Clara, California, USA).

### 3.4. Framework usages

With the networks trained, the first step of adapting the framework to a new wearer $\lambda$ (who may be different from the wearer providing the training data) is to estimate their wearer-specific offsets $\rho^\lambda = (\rho^\lambda_\theta, \rho^\lambda_\phi)$ as introduced in Section 3.2.3. *CalNN* plays an important role in calculating the offsets for new wearers. We collect a set of calibration `fixation points` on the new wearer $\{(g_k, n_k, t_k)|k \in (\text{Samples from } \lambda)\}$, without manually setting any offsets on the companion application. $\{n_k, t_k\}_k$ are used as inputs for the *CalNN*. The wearer-specific $\rho^\lambda$ is estimated as the average difference between $CalNN(n, \dot{t})$ and the angular representation of the captured $\{g_k\}$:

$$(\rho^\lambda_\theta, \rho^\lambda_\phi, *) = \frac{1}{m} \sum_k CalNN(n_k, \dot{t}_k) - \dot{g}_k, \tag{6}$$

See Figure 4($a$) for an illustration. Note that *CalNN* enables us to treat all samples with equal weights, not biasing towards samples near any specific locations like equations (1) and (2).

We then offset the angular representation of $\dot{g}_k$ with the $\rho^\lambda$ value to become $\dot{g}'_k$. *CorNN* then takes $(\dot{g}'_k, n)$ as inputs and outputs a corrected gaze location estimate (see Fig. 4($b$)).

## 4. Evaluation

We evaluated the proposed framework for improving the gaze estimation accuracy. A total of 11 wearers participated in the testing experiment, including `Wearer 0`, who contributed to the training data. The wearers' heads were stabilised on a chin-rest facing the centre of the screen. Unlike the training data collection procedure, the testing experiment used regular monitor screens mounted on a desk: the screen was fixed while the visual target appeared at different locations on the screen, stepping through a grid of ten columns by eight rows sequentially in one recording. The recording is repeated for each wearer at distances 50, 65, 80, 100, and 130 centimetres from the screen, resulting in 400 samples per test wearer. `Wearers 0 to 4` had regular visions without correction and no makeup. The `Wearer 5` had regular vision and wore makeup. The `Wearers 6 to 10` had vision corrections of factor $-2$ or lesser. The `Wearer 10` wore contact lenses during the experiment. We calculated the testing wearers' wearer-specific offsets with the help of the calibration network $CalNN(t, n)$ as explained in Section 3.4. These wearer-specific offsets are then used with the *CorNN* to correct the gaze estimations of the testing wearers. The gaze estimation accuracy for our framework is compared to a baseline estimation method, where the wearer-specific offsets are calculated using Equation (1) and the *CorNN* correction is not applied. Both results are formalised in angular coordinates $(\theta, \phi, d)$. The angular error of an estimated gaze location $\gamma$ with a given ground truth location $\tau$ is:

$$Angular\ Error(\gamma, \tau) = \|(\theta_\tau, \phi_\tau) - (\theta_\gamma, \phi_\gamma)\| \tag{7}$$

In addition to comparing the proposed framework and the baseline method, we have also conducted an ablation study over the proposed network to observe the impact of each component on the whole framework's performance.

## 5. Results

Figure 5 shows the results of absolute angular error from 11 participants. Figure 5(*d*) shows the average accuracy obtained with the baseline method and the proposed correction framework over every testing data. Figure 5(*d*) shows the proposed method has improved the average accuracy of every wearer, with an edge for wearers without vision correction and makeup. It is also noticeable that contact lenses significantly degrade the device's accuracy. `Wearers 1 and 4` benefit from similar improvements to the control test wearer, i.e. `Wearer 0`.

Figure 5(*a*) to 5(*c*) show how the estimation accuracy changes as the distance between the wearer and the screen changes. Figure 5(*a*) shows that the baseline method's accuracy improves with the distance within a short range of $[450, 1300]$ millimetres with improvement factors within about $2 \sim 3\times$. This tendency is also observable in *CorNN*'s results but with a lower impact, between $1 \sim 2\times$ (see Fig. 5(*b*)). Based on Figure 5(*c*), the correction network had overall significantly increased the accuracy at a very short distance $\lesssim 1m$ for most wearers, with an improvement between $0.5°$ and $2°$ and had a positive impact at a longer range for some of them. This result matches our expectation since the lens distortion and parallax effects are most significant at short ranges. Thus, the benefit of applying the networks should be the most prominent. The only exceptions were the accuracy of the

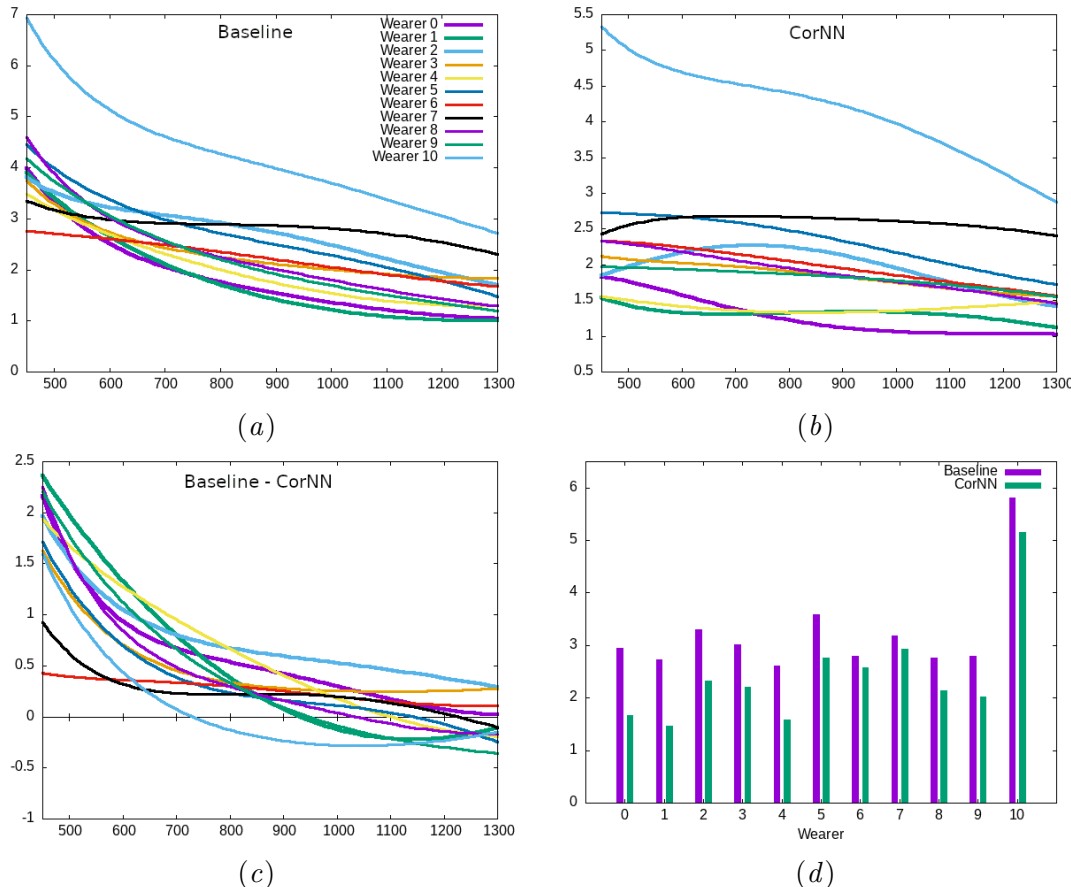

Figure 5: Angular distance between a target and the gaze estimations from the baseline method and the output of the *CorNN* framework. (a) Angular error tendencies of the baseline method. (b) Angular error tendencies of the *CorNN*. (c): Improvement due to the *CorNN* ((a) minus (b)). (d) Average accuracy per wearer. Y-axes of the four plots are angular error values in degree. Figure (a), (b), and (c) x-axes are distance values in millimetres.

`Wearer` 10's measurements at mid-range (around 1m) and several wearers at distances over $1m$. The deterioration varies between $0.1°$ and $0.3°$ approximately.

An ablation study has been conducted to evaluate the impact of each characteristic of the proposed framework introduced in Section 3. The tables 1 and 2 report the average angular error variations upon all testing wearers of the framework with various configurations as a percentage between the original and modified frameworks.

This ablation study showed that, on the one hand, the angular representation, the training data augmentation, and the number of hidden units had the most significant performance impact; on the other hand, the choice of the loss function, the use of the residual and the number of layers contributed half as much to the framework's performance.

| Variable | MAE loss | Cartesian rep. | w/o Residual | w/o Augment. |
|---|---|---|---|---|
| Avg. Ang. Err. | +1.7% | +4.8% | +3.1% | +6.9% |

Table 1: Average angular error variations in percentage, if mean absolute error (MAE) loss was used instead of mean squared error (MSE), Cartesian representation was used instead of the angular representation, the residual was not used, and the augmentation was not used.

| Layers/Hidden units | 500 | 300 | 100 |
|---|---|---|---|
| 20 | X | +2.7% | +6.6% |
| 10 | +1.7% | +3.0% | +6.6% |

Table 2: Average angular error variations in percentage when using 500, 300, or 100 hidden units and 20 or 10 layers (X marks the original configuration).

## 6. Conclusion

The method presented in this paper aims to improve head-mounted eye tracking devices' accuracy by attenuation for the distortions implied essentially by the parallax effect, considering the distance between the observer and the gazed object. This method stands on embedding gaze coordinates into the 3D real-world scene and using two neural networks, one for the correction and one for the calibration. The method has been trained over a single wearer data and tested on eleven wearers upon the Pupil Labs Invisible eye tracking device. The results of those tests have shown that the proposed method could significantly improve the device's accuracy for every tested wearer compared to the baseline method. Further study will include multi-wearer training and few-shot learning techniques to optimise the correction network to its current wearer, integrating the wearer-specific parameters into the neural network architecture.

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
