# OpenReview forum: "Neural Network-Driven Accuracy Enhancement for Wearable Eye-Tracking Devices Using Distance Information"
_NeurIPS.cc/2023/Workshop/Gaze_Meets_ML — NeurIPS 2023 Workshop Gaze Meets ML Withdrawn Submission_

### Official Review · Reviewer_x5s9 · 2023-10-18
**The paper focuses on accuracy enhancement for a mobile eye-tracking device by utilizing two neural networks, CorNN and CalNN using the distance information. The authors show that their method provides accuracy enhancement for all 11 test subjects. However, training was done with only one wearer, so it is a bit open question how generalizable the overall approach is.**

**Rating:** 4
**Confidence:** 3

**Review:**

### Quality
The technical methodology seems valid. However, training was only done with Wearer 0, where this wearer was also included as a test subject in the evaluation. I can imagine collecting a lot of data is demanding and time-consuming, but at least I would have expected a person-independent evaluation here without including Wearer 0 in the test set. The authors also evaluate their method with a baseline but what the baseline is not directly clear at first glance.

### Clarity
Writing should be significantly improved. While proofreading should help in general, Figures 2 and 4 are quite similar, and some parts are redundant. Figure 5 should also be polished with proper legends, axis descriptions, etc. Also, I would encourage authors to report their aggregated results across wearers in their results with mean, std, etc.

### Originality
Related work is quite limited to different applications and settings. How the method addresses the weaknesses of the state-of-the-art is not clear.

### Significance
I think the idea of the authors is good, but there is a lot to improve in the paper.

---

### Official Review · Reviewer_7qHL · 2023-10-23
**The paper entitled "Neural Network-Driven Accuracy Enhancement for Wearable Eye-Tracking Devices Using Distance Information" is addressing and important issue, namely an improvement of gaze estimation in case of wearable eye-tracking  devices.**

**Rating:** 7
**Confidence:** 3

**Review:**

The paper entitled "Neural Network-Driven Accuracy Enhancement for Wearable Eye-Tracking Devices Using Distance Information" is addressing and important issue, namely an improvement of gaze estimation in case of wearable eye-tracking  devices. The approach is important as wearable devices are more challenging to handle than their stationary counterparts. The method is using two networks CorNN and CalNN, respectively.

While the first is meant to correct the bias imposed by the distance between the observer and the locations, the second is meant to improve wearer specific  calibration. The experiments were reported on a private data collection involving 11 subjects.

Altogether, the paper is well written, and properly documented and the method us supported by some experiments, however, there are some issues which should be addressed:
1) Despite the fact that the authors are focusing on a special device with all the specificities of that particular device, it would be appropriate to describe in depth why these issue raise a problem in gaze estimation. More explanation would help the readers to understand the issue at hand.
2) There is a certain lack of motivation in the paper, and somehow the reader got the impressions that the authors did this but not looking much into the current research field.
3) All figures are useful and help much to understand the paper, however Fig. 2 could have been explained more in details.
4) The choice if the network solution is very arbitrary. Why not use something else? It is worth further developing the why part (see the choice of the model!).
5) The data collection and data processing is nicely described. However, the reviewer highly encourages the authors to publish their data, so that others can use it for their experiments. This can also help to compare similar devices from different companies.
6) The data augmentation is not justified and is there a guarantee that all [artificially] generated data is going to represent a possible real life scenario? Please argument.
7) The choice for the networks' structure is completely arbitrary, nothing is specified about the input and output and considering the data possible such large models are not necessary. It would be beneficial to describe properly the model (provide a schematic if necessary) and also explain those "magic numbers" such as the # of hidden units and the dropout rate.
8) The evaluation is somehow limited (see 11 participants), but the reviewer also realizes that such experiment need time and effort to collect substantial amount of data.
9) The results are rather promising, however the results could have been presented in a more obvious way. What is that outlier in Fig. 5a?   Could you please further develop this?!
10) The ablation study, very limited though explains somehow the choice of the networks, but in order to have a proper architecture, more in-depth experiment should be designed.

---

### Official Review · Reviewer_s4rT · 2023-10-24
**The authors propose a technique to improve the accuracy of the wearable eye tracking devices by attenuation of distortions by parallax effect and wearer specific calibration.**

**Rating:** 7
**Confidence:** 4

**Review:**

Purpose:
To improve the accuracy of the wearable eye tracking devices by attenuation of distortions by parallax effect and wearer specific calibration
Significance:
To improve head mounted eye tracking devices accuracy by correcting bias due to parallax effect and lens distortion and wearer specific calibration
Method Proposed:
The authors have used two neural network named as CorNN and CalNN with identical network architectures but have trained them independently. They have generated 10405 samples of training from a single wearer, that consists of gaze location (g), actual target location (t) and camera’s relative orientation to the screen (n). This data was then preprocessed where the Cartesian coordinates were converted to angular coordinates, data was augmented and wearer specific offset was added.
Two networks CorNN and CalNN were trained independently where CorNN takes input n and g to predict target location (t) and CalNN takes input n and t, to predict gaze estimation (g). First calibration of the test set is performed then CorNN is applied to predict the target location thereby correcting the gaze.
The network was tested on 11 participants including the training wearer. 400 samples were generated. 5 participants had regular vision with no makeup, 1 participant had regular vision with makeup. Another 5 had vision corrections and one of these 5 participants wore contact lenses during the experiment. Angular error was used to estimate the accuracy of the final prediction. An ablation study was also conducted on the network by changing the hyperparameters.

Strengths:
-	Deep learning model was applied to improve the accuracy of the wearable eye tracking devices
-	Ablation study provides more information on the required hyperparameters for good results
-	Testing of the data was performed on different types of wearers to get generalizability of the model’s performance
-	Authors provide explanation for the choice of the pre-processing steps

Weaknesses:
-	The results presented in Figure 5 provide comparisons with a baseline method, but no information on the baseline model has been found in the paper.
-	The authors do not provide information on the training hyperparameters like number of epochs, training samples, the amount of augmented data added to the training set and if both networks applied the same training hyperparameters
-	More information can be provided for training of the networks. For example, out of the 10405 samples, if any set of samples were used for validation
-	More information can be provided as to why the wearer’s data that was used for training has also been a part of the testing data
-	More information can be provided for the choice of only using one wearer’s data with regular vision has been applied

Recommendation:
-	To provide comparisons for more than one existing methods that have been applied for improving the accuracy of wearable eye- tracking devices with the proposed model

---

### Meta-Review · Area_Chair_qhhX · 2023-10-26

**Recommendation:** Reject
**Confidence:** 4

**Metareview:**

After analyzing the reviews, it was evident that all the reviewers acknowledged the importance of wearable eye-tracking calibration, but they also mentioned some issues with the paper. They felt that some crucial information needed to be included, and the motivation behind the research needed to be clarified. Additionally, the reviewers felt that it needed to be explained how the proposed methods improved over the existing ones and addressed the current limitations. Specifically, the reviewers were looking for details about the baselines used in the experiment, the hyperparameters for the two proposed networks, and the train-test-validation split. Moreover, the reviewers had an issue with the inclusion of wearer 0 in the training and testing datasets, which biases the test results. They also found that the motivation behind the choice of networks must be clarified despite limited ablation experiments. Finally, the reviewers pointed out that the figures in the paper needed to be more precise.

Despite these issues, all reviewers agreed that the idea behind the research was good, but it required more work to improve the paper. Therefore, the reviewers suggested that the authors consider their feedback and make the necessary improvements to the paper.

---

### Decision · Program_Chairs · 2023-10-26

Reject